# The Moderating Role of Dietary Quality and Dietary Fibre Intake on the Mood Effects of Positive Expressive Writing: A Pilot Study

**DOI:** 10.3390/nu16172875

**Published:** 2024-08-27

**Authors:** Lucie Levová, Michael A. Smith

**Affiliations:** Department of Psychology, Northumbria University, Newcastle upon Tyne NE1 8ST, UK; lucielevova@gmail.com

**Keywords:** dietary quality, dietary fibre, expressive writing, positive affect

## Abstract

Background: Positive expressive writing is associated with enhanced psychological wellbeing. Several individual differences are known to moderate the enhancement effects of positive expressive writing, but no studies to date have investigated the optimal dietary conditions under which expressive writing effects occur. In this pilot study, we sought to investigate whether diet quality and dietary fibre intake moderate the effects of positive writing on mood. Methods: The participants (12 males, 25 females, *M*_age_ = 33.0, *SD*_age_ = 13.1) completed self-reported measures of dietary quality, dietary fibre intake, and positive and negative affect. They were then randomly allocated to complete either a positive expressive writing or neutral writing activity for 10 min. Positive and negative affect were measured again immediately after each activity. Results: Those participants who reported better diet quality and greater dietary fibre intake exhibited a significantly greater increase in positive affect following positive expressive writing relative to neutral writing. No significant effects were observed for negative affect. Conclusions: For the first time, we report that the effects of positive expressive writing on positive mood are enhanced under optimal dietary conditions. Further replication studies are needed to determine whether dietary factors can influence the conditions under which positive expressive writing benefits occur. We speculate that dietary influences on the gut–brain axis are a potential mechanism.

## 1. Introduction

Over the past few decades, several studies have investigated the psychological and physical wellbeing benefits of expressive writing. Expressive writing involves writing about intensely emotional experiences, with a range of different activities proposed from disclosure of trauma to writing about positive life experiences [1]. Expressive writing studies typically require the participants to write (or type) for 15–20 min per day, over three days, with the short-term writing effects investigated on each day of writing and with a follow-up in the weeks or months post-intervention to investigate the persistence of any benefits over the medium term. The expressive writing activity is typically compared to a neutral writing or other control activity. The original expressive writing activity, Written Emotional Disclosure (WED), requires the participants to write about negative emotional experiences or traumas and has been associated with a range of health benefits, including a reduction in self-reported depression and work absenteeism [2]. More recently, many studies have reported the psychological wellbeing benefits of writing about positive topics. In a flagship study, Burton and King (2004) observed that, in comparison to writing about a neutral topic, writing about positive life experiences for 20 min per day, over three consecutive days, was associated with increases in positive mood following each writing session [3].

On this basis, positive expressive writing seems to be beneficial for enhancing positive mood. However, several studies have suggested that these benefits are not universal between individuals, and that it is important to consider factors that may moderate positive expressive writing effects. This is essential for understanding the contextual factors under which positive expressive writing benefits are most reliably observed. Lyubomirsky and Layous (2013) developed the positive activity model, which outlines a range of individual differences, including factors such as motivation, personality, and demographic factors, that are known to influence the efficacy of positive psychology interventions [4]. With respect to positive expressive writing, previous studies have shown that social inhibition moderates the effect of positive expressive writing on trait anxiety [5], depression symptoms, and perceived stress reactivity [1]. It is suggested that individuals who are more socially inhibited may experience greater benefits from engaging with positive expressive writing activities because this provides them with an opportunity for positive expression, which they are unable to express verbally due to their socially inhibited nature [1]. However, on the whole, other psychosocial, lifestyle, or biological factors that may moderate the beneficial effects of positive expressive writing are poorly understood.

It is now well known that a positive relationship exists between dietary quality and psychological wellbeing [6]. Indeed, there is increasing interest in dietary interventions for mental health conditions [7], and it is also understood that dietary quality plays an important role in evoking favourable mental health outcomes [8,9,10]. Studies have found that healthy eating patterns, such as the Mediterranean diet (typically characterised by a high consumption of fruits, vegetables, nuts, and legumes; a moderate consumption of poultry, eggs, and dairy products; and an occasional consumption of red meat), are associated with better mental health and reduced risk of depression [10,11,12,13]. By contrast, unhealthy eating patterns, such as the Western diet (typically consisting of a low intake of fruits and vegetables and a considerably high consumption of animal-derived protein, saturated fats, refined grains, sugar, salt, and alcohol), are believed to be detrimental to peoples’ mental health [10,14]. Further, it has been established that the intake of specific macronutrients, including dietary fibre, confers positive mental health benefits [15,16].

Recently, studies have begun to investigate the mechanisms underpinning these dietary influences on psychological wellbeing. The dietary intake and habits can modify the gut microbiome composition—a diverse community of trillions of microbes, bacteria, viruses, fungi, and protozoa that all significantly contribute to metabolism, preventing the development of Western diseases, and extending the lifespan [17]. It has been found that the brain and the gut communicate bidirectionally through the gut–brain axis [18,19], meaning that any disruption in the gut may lead to poor mental health and vice versa. It has previously been suggested that dietary approaches targeting microbiota specifically, to positively modify gut–brain communication, could be used in the future to reduce perceived stress in the human population [20]. Studies have indeed implied that the gut microbiome plays a very important role in mental health and psychological wellbeing [21]. However, the Western dietary pattern can have considerably negative effects on the composition, diversity, and function of the microbiome composition [22,23,24]. By contrast, one of the key factors that has also been found to help maintain a healthy and balanced gut is an increased consumption of dietary fibre (which can be found in foods such as fruits, vegetables, legumes, or whole grains, for instance). In turn, this is a potential mechanism via which dietary fibre intake can have beneficial effects on mental health [25,26]. 

It has been established above that optimising the dietary quality and increasing the fibre intake can confer benefits for wellbeing. Positive expressive writing is also known to increase positive mood [3], and to reduce stress and anxiety [5]. However, it is known that a range of contextual and inter-individual factors likely moderate the effects of positive expressive writing [4], and at present these are poorly understood. Given that the dietary quality and fibre intake, potentially mediated by the influences of dietary fibre on the gut–brain axis mechanisms, can enhance psychological wellbeing, it is possible that good dietary quality may provide the optimal conditions for psychological interventions, such as positive expressive writing, to maximally influence psychological wellbeing. The idea that better dietary quality can enhance the efficacy of interventions is not new. For example, Young and colleagues (2022) [27] observed that micronutrient supplementation improved the performance on an attention task, but only in individuals with an ‘optimal’ diet pre-supplementation. Therefore, the aim of the study was to investigate the moderating role of dietary quality and fibre intake on the association between a single session of positive expressive writing and mood. The pre-registered hypothesis (https://osf.io/9zqvg, accessed on 15 August 2023) was that there will be a significant positive effect of positive expressive writing on psychological wellbeing. Given the novel aim to explore the moderating role of dietary fibre and dietary quality, this aim is exploratory, although, speculatively, we suggest that any mood effects of positive writing will be stronger in individuals with better dietary quality and a higher self-reported dietary fibre intake.

## 2. Materials and Methods

### 2.1. Participants

Data were collected online via Qualtrics software (www.qualtrics.com, accessed on 20 March 2023). The study was conducted in accordance with the Declaration of Helsinki, and the protocol was approved by the Ethics Approval Process of Northumbria University (reference 3719). The study protocol was pre-registered on the Open Science Framework (https://osf.io/9zqvg, accessed on 15 August 2023). Recruitment began on 13 April 2023 and terminated on 13 August 2023. Participants were recruited through an opportunity sampling methodology. Online advertisements were placed on various social media platforms, including LinkedIn, Instagram, Facebook, and Reddit, with a link to access the study, provide informed consent, and complete the study online. Exclusion criteria were a diagnosis of any gut health conditions and/or digestive disorders, as well as a previous or current diagnosis of any type of eating disorder. Participants were required to be fluent in written English. Prior to conducting the research, a power analysis was conducted on G*Power [28] with an alpha level of 0.05 and power of 0.80. This indicated that a minimum of 55 participants would be required to obtain a medium effect.

### 2.2. Materials

#### 2.2.1. Positive and Negative Affect Schedule (PANAS) [29]

The PANAS questionnaire was used in the present study to indicate individual levels of subjective psychological wellbeing. This tool comprises two brief 10-item mood scales that assess positive and negative affect. Positive affect (PA) descriptors include feeling ‘interested’, ‘excited’, ‘strong’, ‘enthusiastic’, ‘proud’, ‘alert’, ‘inspired’, ‘determined’, ‘attentive’, and ‘active’. Negative affect (NA) descriptors include feeling ‘distressed’, ‘upset’, ‘guilty’, ‘scared’, ‘hostile’, ‘irritable’, ‘ashamed’, ‘nervous’, ‘jittery’, and ‘afraid’. Participants respond on a 5-point Likert scale (varying from “1—very slightly to not at all” to “5—extremely”), with a total score being computed for each subscale. Participants were asked to ‘indicate the extent to which you feel this way in the present moment’; this was to ensure that PANAS was effectively used to measure current state mood. The internal reliability and construct validity of the PANAS have been successfully established previously [30]. However, as the instructions for completing the PANAS questionnaire were modified in the present study for the purpose of measuring current state mood, internal reliability was computed to confirm acceptable psychometric properties in the present study sample. The pre-intervention PANAS showed good reliability of the PA scale (α = 0.81) and the NA scale (α = 0.87), and the post-intervention PANAS showed excellent reliability of the PA scale (α = 0.90) and the NA scale (α = 0.91).

#### 2.2.2. Dietary Fibre Intake Food Frequency Questionnaire (DFI-FFQ) [31] 

The DFI-FFQ questionnaire was used in the present study as a measure of dietary fibre intake (DFI). It composed of a total of five items, each requiring participants to indicate their past year average intake of foods rich in fibre (fruits, vegetables, breads and cereals, nuts and seeds, and legumes). Each item comprised a 12-point response scale, requiring participants to indicate their average consumption (number of serves) from ‘never’ to ‘6 or more serves per day’. Examples were provided of what constitutes a single serve. For the purpose of the present study, scores on each item were summed to derive a total score. Higher scores indicated greater DFI. The DFI-FFQ has been previously classified as a valid tool for measuring DFI and an accurate measure for classifying individuals based on habitual DFI, particularly in New Zealand population [32].

#### 2.2.3. Short-Form Food Frequency Questionnaire (SFFFQ) [33]

The SFFFQ was used as a measure of typical diet composition and quality. For the purposes of the present study, only the ‘food and drink intake’ questions were used. Participants were asked about the frequency during a ‘typical’ week over the past month with which they consumed one portion of 20 foods that are typically consumed as part of a mainstream British diet. Participants responded on an 8-point scale from ‘rarely or never’ to ‘5+ a day’. The scoring tool that accompanies this measure was used to derive a total dietary quality score (DQS) for each participant, which was used in the main analyses. Higher DQS indicates better dietary quality and is derived by summing the dietary intake component scores for fruits (1 = ≤2 servings/week; 2 = >2 servings per week; <2 servings per day; 3 = ≥2 servings per day), vegetables (1 = ≤1 servings/day; 2 = 1–3 servings per day; 3 = ≥3 servings per day), oily fish (1 = no intake; 2 = 0–200 g per week; 3 = ≥200 g per week), fat (1 = ≥1.5 times UK recommendations; 2 = 1–1.5 times UK recommendations; 3 = ≤UK recommendations), and non-milk extrinsic sugars (NMES; 1 = ≥1.5 times UK recommendations; 2 = 1–1.5 times UK recommendations; 3 = ≤UK recommendations). The SSFFQ calculations are based on UK-recommended fat and NMES intakes of 85 g/day and 60 g/day, respectively. The SFFFQ authors purport that the questionnaire can be used as a suitable tool for assessing diet quality within the UK population [33]. It has been successfully used within numerous studies assessing dietary quality in various populations [34,35,36]. 

#### 2.2.4. Linguistic Inquiry and Word Count (LIWC) Software [37]

Linguistic Inquiry and Word Count (LIWC)-22 software [37] was used to analyse the use of positive emotion words in the writing tasks. The software has been standardised across a large range of text samples, derived from different contexts and authors [38], and has been used in previous expressive writing activities to measure the frequency of emotion word use. In the present study, LIWC was used as a manipulation check to ensure participants randomised to the positive expressive writing condition used a greater frequency of positive emotion words. 

### 2.3. Procedure

All participants provided their informed consent for inclusion before they participated in the study. Participants were able to complete the study on any electronic device with an internet connection, on which they were able to type responses. As all data were collected online, participants were able to complete the study at a time and place convenient to them.

Participants were first asked to provide their age and gender. They then completed the SFFFQ and DFI-FFQ before completing the PANAS. Participants were then randomised to complete either a positive expressive writing activity or a neutral control writing activity using the Qualtrics randomiser function. The ‘Evenly Present Elements’ option was not used; thus, the number of participants previously allocated to a condition was not taken into account when randomising subsequent participants. The instructions for the positive writing activity were obtained from Burton and King (2004) [3]. Participants were instructed to “Think of the most wonderful experience or experiences in your life, happiest moments, ecstatic moments, moments of rapture, perhaps from being in love, or from listening to music, or suddenly ‘being hit’ by a book or painting or from some great creative moment. Choose one such experience or moment. Try to imagine yourself at that moment, including all the feelings and emotions associated with the experience. Now write about the experience in as much detail as possible trying to include the feelings, thoughts, and emotions that were present at the time. Please try your best to re-experience the emotions involved”. The instructions for the neutral control writing activity were obtained from Hansen et al. (2022) [39]. Participants were instructed to “Describe what you did yesterday in as much detail as possible, from the moment you woke up to the time you went to bed. Include as many details from the days as possible. We’re less interested in the emotions associated with the activities you undertook, and more interested in a detailed, factual account of what you did”. In both conditions, participants were instructed to write for at least 10 min and were not able to advance past the writing activity until 10 min had elapsed. Participants were not informed about the true aims of the study, the known benefits of positive expressive writing, or that there were two writing conditions. They were informed only that they would be required to write for 10 min about “past experiences”. Following the writing activity, both groups completed the PANAS questionnaire once more. The study procedure is summarised in Figure 1.

### 2.4. Design and Analysis

The analysis plan was pre-registered (https://osf.io/9zqvg, accessed on 15 August 2023) prior to the commencement of data analysis. A between-subjects design was employed. Change scores were computed for positive affect and negative affect by subtracting the pre-writing values from the post-writing values. Data were screened for outliers, with the intention to treat as missing any extreme outlying values (M ± 3SD). Skewness and kurtosis values were computed in SPSS for all study variables. Independent samples *t*-tests were run on baseline positive affect, negative affect, and self-reported dietary intake variables to ensure no significant baseline differences between participants randomised to the two writing conditions. A manipulation check was performed via an independent samples *t*-test, comparing the frequency of positive tone words used, derived via LIWC, between the two conditions.

The moderation analyses were run using the PROCESS macro version 4.2 for SPSS version 28 [40]. Four moderation analyses were run, with condition (sum-coded; positive expressive writing = 1; control writing = −1) entered as the IV for all analyses. The first two analyses included positive affect change as the DV, with dietary quality and dietary fibre intake entered as moderators in respective analyses. The final two analyses repeated this procedure, with the exception that negative affect change was included as the DV. Variables were mean-centred and conditioning values were set to −1SD, mean, +1 SD. Simple slopes were plotted for visualisation of significant interaction effects. The data are available via the Open Science Framework (https://osf.io/akfer, accessed on 15 August 2023). At the request of an independent reviewer, each moderation analysis was re-run, with the other diet variable included as a control variable.

## 3. Results

In total, sixty participants clicked the link to take part in the study and reached the consent question. Of those, five participants did not consent to take part in the study, and eighteen people did not complete the entire study, resulting in a final 38% overall attrition rate. Therefore, the final sample for the analysis included thirty-seven participants (*M*_age_ = 33, *SD*_age_ = 13.08) and comprised twenty-five females (*M*_age_ = 30.96, *SD*_age_ = 11.89) and twelve males (*M*_age_ = 37.25, *SD*_age_ = 14.92).

Of the final thirty-seven participants included in the analysis, fourteen were in the positive writing group (*M*_age_ = 35.07, *SD*_age_ = 14.74), of which ten were females and four were males. Correspondingly, twenty-three participants were in the neutral control group (*M*_age_ = 31.74, *SD*_age_ = 12.13), incorporating fifteen females and eight males. 

No extreme outliers (*M* ± 3 *SD*) were detected. All the variables included in the analyses showed values in the acceptable range (range = −0.69–0.16). The kurtosis value for positive affect (2.81) suggested a mildly leptokurtic distribution. The kurtosis values for the other variables were acceptable.

### 3.1. Baseline Data

The participants’ self-reported dietary intake of fruits, vegetables, oily fish, fat, and NMES, overall diet quality scores, dietary fibre intake FFQ scores, and baseline PANAS scores are shown in Table 1. There were no significant differences on any baseline variables between the participants randomised to the two writing conditions. There were no statistically significant correlations between the baseline positive affect, negative affect, and the diet variables.

### 3.2. Manipulation Check

The participants in the positive expressive writing condition (*M* = 7.00, *SD* = 3.15) used a significantly greater frequency of words with a positive tone relative to the neutral control writing condition (*M* = 3.98, *SD* = 3.01), *t* (35) = −2.92, *p* = 0.006, *d* = −0.99. This suggests that, on average, the participants in each condition adhered to the instructions of their relative activity. 

### 3.3. Positive Affect

For dietary quality, the overall model was significant, *F* (3, 33) = 3.56, *p* = 0.024, *R*^2^ = 0.24. The effects of the condition, *b* = 3.49, *SE* = 2.01, *t* = 1.73, *p* = 0.092, and dietary quality, *b* = 1.28, *SE* = 0.68, *t* = 1.87, *p* = 0.071), were nonsignificant. The interaction between the condition and dietary quality was significant, *b* = 3.84, *SE* = 1.54, *t* = 2.49, *p* = 0.018). The analysis of the simple slopes revealed that positive expressive writing was significantly associated with a greater increase in positive affect, but only at good (+1 SD), *b* = 9.44, *SE* = 3.08, *t* = 3.07, *p* = 0.004, but not at moderate, *b* = 3.49, *SE* = 2.01, *t* = 1.73, *p* = 0.093), or poor (+1 SD), *b* = −2.467, *SE* = 3.18, *t* = −0.78, *p* = 0.443), levels of dietary quality (see Figure 2). These effects remained when dietary fibre intake was included in the analysis as a covariate.

For dietary fibre intake, the overall model was significant, *F* (3, 33) = 4.45, *p* = 0.010, *R*^2^ = 0.29. The effects of the condition, *b* = 4.48, *SE* = 1.97, *t* = 2.27, *p* = 0.030, and dietary fibre intake, *b* = 0.31, *SE* = 0.15, *t* = 2.06, *p* = 0.047, were significant. Further, the interaction between the condition and dietary fibre intake was significant, *b* = 0.70, *SE* = 0.31, *t* = 2.30, *p* = 0.028. The analysis of the simple slopes revealed that positive expressive writing was significantly associated with a greater increase in positive affect, but only at good (+1 SD), *b* = 9.05, *SE* = 2.88, *t* = 3.15, *p* = 0.004, or moderate, *b* = 4.48, *SE* = 1.97, *t* = 2.27, *p* = 0.030), levels of dietary fibre intake. The slope for low levels of dietary fibre intake, *b* = −0.01, *SE* = 2.71, *t* = −0.04, *p* = 0.971, was nonsignificant (see Figure 3). These effects remained when dietary quality was included in the analysis as a covariate.

### 3.4. Negative Affect

For dietary quality, the overall model was nonsignificant, *F* (3, 33) = 0.48, *p* = 0.700, *R*^2^ = 0.04. The effects of the condition, *b* = 0.79, *SE* = 1.17, *t* = 0.68, *p* = 0.504, dietary quality, *b* = −0.27, *SE* = 0.40, *t* = −0.68, *p* = 0.499, and the condition x dietary quality interaction, *b* = 0.39, *SE* = 0.90, *t* = 0.43, *p* = 0.668, were nonsignificant. For dietary fibre intake, the overall model was nonsignificant, *F* (3, 33) = 0.27, *p* = 0.843, *R*^2^ = 0.04. The effects of the condition, *b* = 0.79, *SE* = 1.17, *t* = 0.68, *p* = 0.504, dietary fibre intake, *b* = −0.27, *SE* = 0.40, *t* = −0.68, *p* = 0.499, and the condition × dietary fibre intake interaction, *b* = 0.39, *SE* = 0.90, *t* = 0.43, *p* = 0.668, were nonsignificant.

## 4. Discussion

The primary aim of the present study was to investigate the possible moderating role of dietary quality and dietary fibre intake on the psychological wellbeing benefits of positive expressive writing. The key findings were that both dietary quality and dietary fibre intake moderated the effects of positive expressive writing on positive affect. Specifically, a significant increase in positive affect was observed following positive expressive writing only at high levels of dietary quality, and at high and moderate but not low levels of dietary fibre intake. To the best of our knowledge, this is the first report in the literature that diet can influence the efficacy of expressive writing activities. However, given the small sample size, this should be considered as a pilot study, and further work is needed to replicate these effects in a larger sample. There were no significant effects observed for changes in negative affect. 

Several studies have reported that unhealthy eating patterns (such as the Western diet) may be detrimental to mental health [10,14]. On this basis, we were particularly interested here to investigate the moderating role on positive expressive writing effects of self-reported dietary quality. Further, given the purported benefits of a healthy and balanced gut microbiome for mental health [25,26] and the known influence of dietary fibre intake on gut health [41], we wanted to explore the moderating role of self-reported dietary fibre intake on positive expressive writing effects. Previous studies have found that psychological moderators, including social inhibition [1,5] and alexithymia [42], moderate the effects of positive expressive writing on the psychological outcomes. In the present pilot study, we were interested to extend our understanding of those non-psychological factors that potentially moderate positive expressive writing effects. Our exploratory prediction that dietary quality and dietary fibre intake would enhance the effect of positive expressive writing on positive affect was speculative owing to the fact that no previous studies have investigated expressive writing benefits in the context of dietary quality. However, our aims were predicated on previous nutrition intervention studies that suggest that nutritional interventions may confer optimal benefits in individuals with good dietary quality. For example, Young and colleagues (2022) observed that a multivitamin supplement intervention was effective for improving the performance on an attention task, but only in individuals with an optimal diet at the baseline [27]. We were interested here to explore whether this notion could be extended to psychological interventions, in this case expressive writing. Our findings suggest that expressive writing interventions are most likely to be effective in individuals with better diet quality. Speculatively, this could be due to dietary influences on the gut microbiome and gut–brain axis, a biological mechanism for the optimal conditions under which expressive writing benefits may occur. It is known that fruit, vegetable, and fibre intake can increase the number of beneficial gut microbiota and decrease harmful ones. Gut microbiota are known to influence mental health via a number of mechanisms, including the regulation of the production of neurotransmitters such as gamma-aminobutyric acid (GABA) and serotonin as well as amino acids including tryptophan. Gut microbiota also play a role in the functioning of the immune system, which in turn can influence the key neuroendocrine pathways involved in maintaining mental health, such as the hypothalamic–pituitary–adrenal (HPA) axis [43]. Given the robust influence of diet on mental health, one implication may be that no intervention may be strong enough to compensate for the adverse consequences of a poor diet. Should the present study findings be replicated, the positive activity model [4] could be expanded to incorporate the moderating role of diet on positive expressive writing efficacy. 

It is noteworthy that, in the present study, no main effects of the writing condition on changes in either positive or negative affect were observed. This contradicts the previous studies that have demonstrated that positive expressive writing can enhance positive mood [3] and reduce stress and anxiety [5]. A key difference between this previous work and the present study is that, previously, the participants were instructed to write for 20 min, whereas the participants in the present study wrote for only 10 min. It could be concluded that 10 min is not a sufficient writing duration to induce mood effects. However, it has been demonstrated previously that two minutes of positive writing is sufficient to induce a change in positive mood [44]. Another difference between these studies is the mode of writing. The present study required the participants to type, whereas the participants used handwriting in previous studies that have observed significant mood effects [3,5]. However, a further previous study that required the participants to type [1] also observed only moderation effects of social inhibition but no main effects of condition. Therefore, it could be concluded that typing is a less sensitive mode of writing than handwriting, whereby effects are only observed in the most optimal contexts. This point notwithstanding, it is important to note that the present study was likely underpowered. The a priori power analysis indicated that a minimum sample size of 55 would be required to observe medium effects. However, despite initially recruiting 60 participants to take part in the study, only 37 participants were included in the analysis given the high attrition rate of the present study. This might also account for the lack of main effects. Although this high (38%) attrition rate contributed to the study being underpowered, it is not unusual to see a high dropout rate in expressive writing studies. Indeed, the present study dropout rate is similar to the 36% dropout rate reported by Round and colleagues (2022) for a positive expressive writing study [45]. The high dropout rate in studies investigating these kinds of activities could also be viewed in the context of Lyubomirsky and Layous’ (2013) positive activity model, which suggests that individual differences dictate how suitable any intervention may be for a given individual [4]. There are likely some people who do not like or do not feel comfortable writing expressively about emotions, and such individuals are more likely to discontinue their participation.

A further limitation of the present study was the use of subjective self-report measures of dietary quality and fibre intake. Due to social desirability bias, the participants may have consciously or subconsciously wanted to convey healthier dietary habits when self-reporting their dietary intake. This could have been overcome by employing a food diary alongside an objective measure, such as having the participants take photos or videos of all the food consumed to verify the precise foods consumed and portion sizes [46]. Alternatively, the nutrient status could be measured via blood biomarkers. Similarly, the reported data might have been subject to recall bias or under/over-reporting as FFQs tend to be heavily reliant on peoples’ memory and ability to accurately estimate portion sizes [47]. No data were collected on nationality, and it is plausible that participants from a range of countries, with different dietary patterns that are not accounted for by the questionnaires employed here, participated. Additionally, in the present study, we included no measures of socioeconomic status (SES). There have been calls for expressive writing studies to include measures of SES [48,49] as, speculatively, better-educated, higher-SES individuals may be better engaged and experience greater benefits from expressive writing. Further, there is a known relationship between SES and diet quality [50]. It is plausible that measures of dietary quality and dietary fibre intake are essentially proxy measures for SES, and this could account for the significant moderation effects observed in the present study. Generally, a limitation of the study is that only minimal demographic information was obtained from the participants, which makes it difficult to fully characterise the study sample. Additionally, the cross-sectional study design means that the influence of the dietary intake over time or the longevity of the expressive writing benefits could not be determined here.

## 5. Conclusions

In summary, we have explored whether dietary quality and dietary fibre intake moderate the effect of positive expressive writing on changes in mood. For the first time, we report that dietary factors moderate these effects. Specifically, positive expressive writing induced a significant increase in positive mood relative to neutral writing, but only in individuals with better self-reported dietary quality and higher dietary fibre intake. However, given the attrition and small sample size, the present investigation should be considered a pilot study. Further replication studies with substantially larger samples are needed to ascertain whether dietary factors can optimise the conditions under which positive expressive writing benefits occur before conclusive inferences can be drawn from this evidence. Speculatively, we suggest that dietary influences on the gut microbiome, and consequently the gut–brain axis, may be a mechanism via which these effects arise. However, it is possible that the self-reported dietary factors here are merely a proxy for other unmeasured mechanisms, such as SES.

## Figures and Tables

**Figure 1 nutrients-16-02875-f001:**
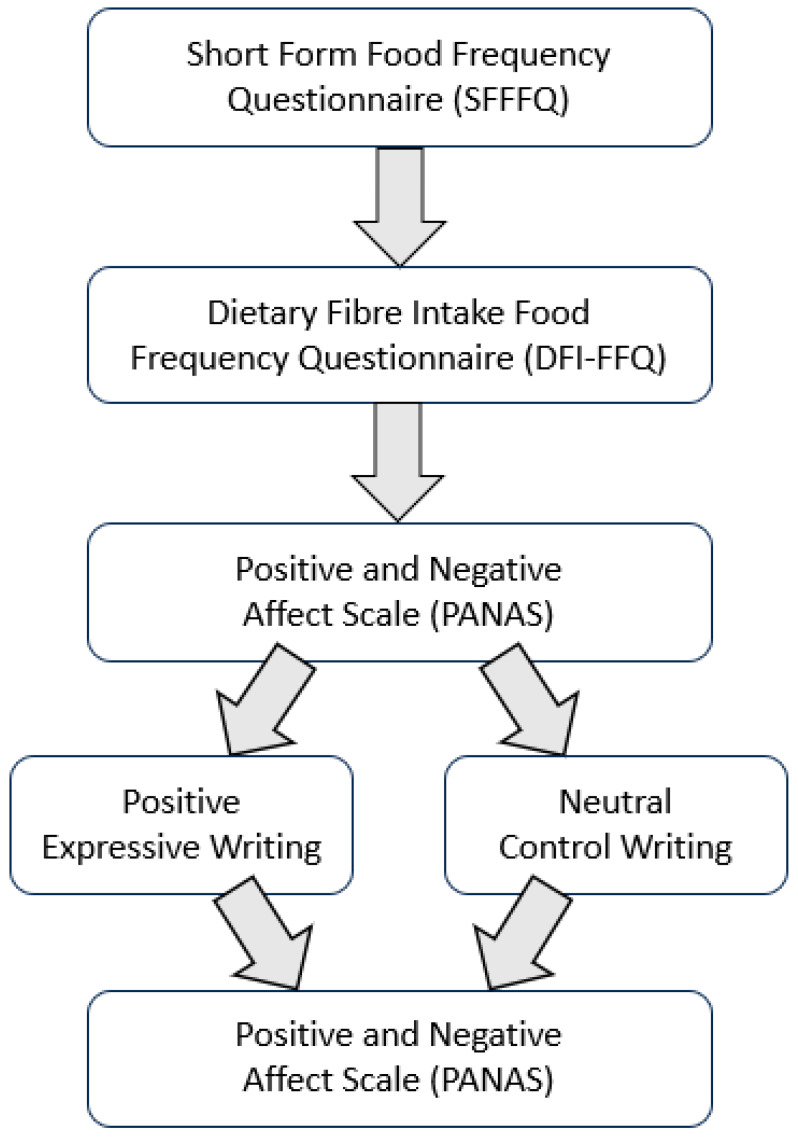
Outline of the study procedures.

**Figure 2 nutrients-16-02875-f002:**
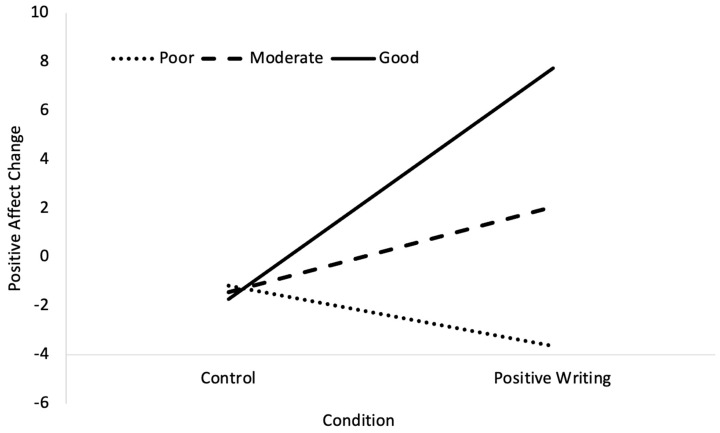
Simple slopes for the relationship between condition and positive affect change, moderated by dietary quality. At good (+1 SD) dietary quality, the relationship was significant. The relationship was nonsignificant at moderate (mean) and poor (−1 SD) dietary quality.

**Figure 3 nutrients-16-02875-f003:**
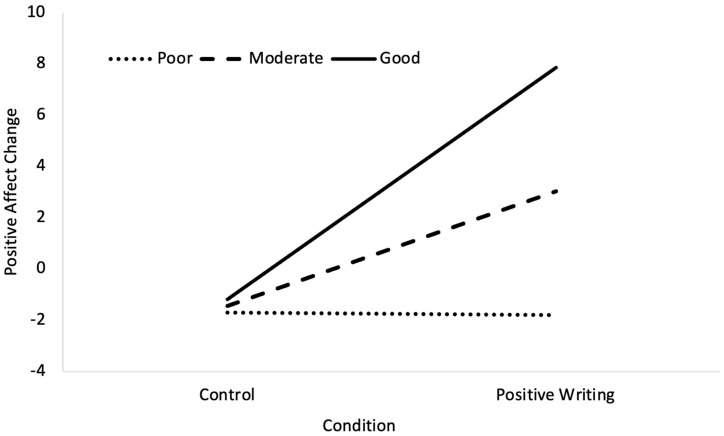
Simple slopes for the relationship between condition and positive affect change, moderated by dietary fibre intake. At good (+1 SD) and moderate (mean) levels of dietary fibre intake, the relationship was significant. The relationship was nonsignificant at poor (−1 SD) dietary quality.

**Table 1 nutrients-16-02875-t001:** Self-reported baseline PANAS and dietary intake variables. There were no significant differences on any baseline variables between participants randomised to the two writing conditions.

Variable	Whole Sample		Positive Writing		Neutral Writing		*p*
	*M*	*SD*	*M*	*SD*	*M*	*SD*	
Positive Affect	29.0	5.9	27.5	5.4	30.0	6.2	0.22
Negative Affect	16.6	6.3	17.4	7.8	16.1	5.2	0.54
Fruits	1.9	0.6	1.9	0.6	1.9	0.6	0.94
Vegetables	1.7	0.8	1.6	0.7	1.7	0.8	0.72
Oily Fish	1.8	0.6	1.7	0.7	1.8	0.6	0.61
Fat	2.5	0.6	2.5	0.7	2.6	0.5	0.82
NMES	2.7	0.6	2.8	0.6	2.6	0.6	0.53
Overall Dietary Quality	10.6	1.5	10.7	1.2	10.6	1.7	0.84
Dietary Fibre Intake FFQ score	28.3	6.5	27.3	6.7	28.9	6.5	0.48

## Data Availability

Data are publicly available via the Open Science Framework (https://osf.io/akfer, accessed on 15 August 2023).

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
