# Peer review of "The Moderating Role of Dietary Quality and Dietary Fibre Intake on the Mood Effects of Positive Expressive Writing: A Pilot Study"

_nutrients, 2024, doi:10.3390/nu16172875_

Round 1

Reviewer 1 Report (Previous Reviewer 3)

Comments and Suggestions for Authors

The study conducted by Levová & Smith investigated the diet quality and dietary fiber intake on the effects of positive writing on mood. The study is original, has a small n, and should undergo some adjustments.

At the beginning of the material and methods, under “participants”, it is interesting to start the paragraph describing the place where the study was conducted, the approval by the ethics committee with the registration number, the clinical trial registration, followed by the eligibility criteria and ending with the sample calculation. In addition, it is important to describe how the randomization was carried out and whether the study was blinded. I suggest that the authors follow the Consort Guidelines;

It would be interesting to perform an intention-to-treat analysis for the lost sample;

In addition to the analyses presented, a correlation analysis in positive affect and diet would reinforce the results;

It is important for the authors to discuss which physiological mechanisms between the gut-brain axis are activated in the results found. They can highlight biological and cellular mechanisms at the brain level.

Author Response

Please see attached file detailing the responses to all reviewer comments.

Reviewer 2 Report (New Reviewer)

Comments and Suggestions for Authors

The purpose of this pilot study was to evaluate the cross-sectional relationship between dietary quality and dietary fiber intake on mood effects and positive expressive writing. The authors provide sufficient rationale to conduct this would with an appropriate background of the literature. My major concern is the lack of a sufficient description of the subjects with no consideration for how some subject characteristics may influence outcome measures, e.g., education, social economic status (although this is mentioned in the Discussion), alcohol intake, medications, age, sex, healthy?. These are my additional comments:

1.      It would have been nice to see how tease out if the relationships with positive affect of dietary fiber and dietary quality? Could this have been done by adjusting for intake of one while analyzing for the other. For example, could it be that the relationship with fiber due to it being a biomarker of plant food intake? Could there have been another plant food component accounting for the relationship with plant foods, e.g. polyphenols (which are prebiotics)?

2.      There is no description of the environment in which the assessments were taken. Was this in the home? In the research institution? Time of day? Time after a meal? Could these have been variables to affect the outcome?

3.      The dropout was quite high (38%). Although the authors address this in the Discussion there is concern that this could have presented a bias to the study results. That is, were the dropout subjects different in any way from those that completely participated in the study?

4.      A better study design would have been to have this be a cross-over study. This would have helped control for variability in subject characteristics and would have been easy to do.

5.      A limitation to this study is that it is cross-sectional. This should be stated.

Author Response

Please see attached file detailing the responses to all reviewer comments.

This manuscript is a resubmission of an earlier submission. The following is a list of the peer review reports and author responses from that submission.

Round 1

Reviewer 1 Report

Comments and Suggestions for Authors

nutrients-2784335-peer-review-v1.writing.review

2.1. Participants

“Prior to conducting the research, a power analysis was conducted on G*Power (Faul 118 et al., 2009) with an alpha level of .05 and power of .80. This indicated that a minimum of 55 participants would be required to obtain a medium effect. Participants were recruited through an opportunity sampling methodology. Online advertisements were placed on various social media platforms, including LinkedIn, Instagram, Facebook, and Reddit, with a link to access the study, provide informed consent, and complete the study online. Some specific exclusion criteria applied due to the nature of the study. Participants were required to be fluent in written English. Additionally, participants with any gut health 125 conditions and/or digestive disorders were asked to not take part, as well as individuals with a previous or current experience of any type of eating disorder. “

3. Results

“In total, 60 participants clicked the link to take part in the study and reached the consent question. Of those, five participants did not consent to take part in the study, and 18 people did not complete the entire study, resulting in a final 38% overall attrition rate. Therefore, the final sample for analysis included 37 participants (Mage = 33, SDage = 13.08) and comprised 25 females (Mage = 30.96, SDage = 11.89) and 12 males (Mage = 37.25, SDage = 14.92).

Of the final 37 participants included in the analysis, 14 were in the positive writing group (Mage = 35.07, SDage = 14.74), of which 10 were females and four were males. Correspondingly, 23 participants were in the neutral control group (Mage = 31.74, SDage = 12.13), incorporating 15 females and eight males.”

The study aimed for 55 people, but only got 37 people, with 14 in the positive writing group (10F) and 23 in the neutral control group (15F). Thus, the study focused on 14 people, mostly female, which is much too small.

Discussion

“It is noteworthy that in the present study, no main effects of writing condition on changes in either positive or negative affect were observed. This contradicts previous studies which have demonstrated that positive expressive writing can enhance positive mood (Burton & King, 2004) and reduce stress and anxiety (Smith et al, 2018). “

The procedure did not work, perhaps because the expressive writing time was too short as the authors suggest.

“A further limitation of the present study was the use of subjective self-report measures of dietary quality and fibre intake. Due to social desirability bias, participants may have consciously or unconsciously wanted to convey healthier dietary habits when self-reporting their dietary intake. Similarly, the reported data might have been subject to recall bias or under/over-reporting, as FFQs tend to be heavily reliant on peoples’ memory 357 and abilities to accurately estimate portion sizes (Hooson et al., 2022). “

The authors could have also added another measure to check the self -report of diet.

“Further replication studies with substantially larger samples are needed to ascertain whether 374 dietary factors can optimise the conditions under which positive expressive writing benefits occur.”

Correct; this study can only be viewed as a pilot study. not a full study, and the results are not worth reporting at this stage.

Reviewer 2 Report

Comments and Suggestions for Authors

I read the manuscript titled 'Positive expressive writing has greater benefits for positive affect in individuals reporting better dietary quality and higher dietary fibre intake' with great interest due to the subject matter and interesting study design. However, the study group size was too small to allow for the expected effects to occur, despite the promising results. The authors correctly identify a significant limitation of their study and suggest that it should be repeated with a larger group of subjects. I fully agree with this assessment. Although the study design is interesting and the premise is valid, the small sample size prevents reliable conclusions. Nevertheless, the results are noteworthy and merit publication, and the study should be pursued further.

Another serious limitation of the study, pointed out by the authors, is the failure to take into account the influence of other factors on the results obtained, such as the socio-economic characteristics, nationality and eating habits of the subjects. In addition, I believe that the analysis of the results obtained should also take into account the individual characteristics of the subjects and related to their ability to express in writing their feelings, thoughts, emotions, etc., which in my opinion can influence the results of the study to a very significant degree. All these factors were identified by the authors of the manuscript, but for some reason were not included in the data analysis.

Below are my comments on the reviewed manuscript:

- the Abstract section should be supplemented with a brief description of the study group,

- the Introduction section needs to be shortened and key points left out, especially as much of the content of this section is repeated in the Discussion section,

- throughout the manuscript, the insertion of references needs to be improved - it should be adapted to the editorial requirements of the journal,

- the Materials and Methods section should be supplemented with the date of the study, the place where the study was conducted, the full version of the inclusion and exclusion criteria,

- regarding the procedure of the study (section 2.3), the question arises whether the authors took into account the possibility that "neutral control writing" could overlap with "positive expressing writing"? And vice versa? Some subjects (depending on individual characteristics) may have just had an expressive writing style and were unable to write a neutral text. A similar difficulty may have occurred in the other direction, i.e. individuals may not have had complete expressive writing skills. In addition, it may have been difficult for the subjects to write about their emotions at all

- Figure 1 needs an explanation of the stages of the study and an explanation of the abbreviations. The current version contains only abbreviations, which may mean absolutely nothing to a person who has not read the text of the manuscript

- in the Results section, there is very little characterisation of the study group; in fact, very little is known about the people who took part in the study

- The Results section includes a very small number of results. I believe it would have been useful to include results from the SFFFQ and DFI-FFQ questionnaires, which would have made it possible to present the characteristics of the study group in terms of frequency of intake of individual product groups, declared dietary fibre intake and diet quality. I believe that from the point of view of the purpose of the study, such characterisation should be included in the manuscript

- the References section needs to be improved and adapted to the editorial requirements of the journal, e.g. the insertion of the Abbreviated Journal Name is required, the year of publication should be bolded and inserted after the title of the manuscript.

Reviewer 3 Report

Comments and Suggestions for Authors

The study conducted by is an original work, well designed and with appropriate methodological approaches. The study has an indication of acceptance, but with some pending issues that, when corrected or answered, could improve it.

- The authors had a considerable sample loss; however, they chose not to use intention-to-treat analysis. What was the reason?

-The authors must describe more clearly how the study randomization was carried out, in addition to the primary and secondary outcomes and eligibility criteria;

- In the discussion, more studies are needed to support the main result of the study.

Round 2

Reviewer 1 Report

Comments and Suggestions for Authors

In response to my review below (and perhaps other reviews), the authors have added that this is a pilot study. However, in most cases pilot studies are not published; they lead to larger-scale studies which are published.  Because the sample here is so small and so biased (female, young) there is a high probability that the results will not be replicable.These data will confuse/mislead the field, not help the field.

From original review:

Of the final 37 participants included in the analysis, 14 were in the positive writing group (Mage = 35.07, SDage = 14.74), of which 10 were females and four were males. Correspondingly, 23 participants were in the neutral control group (Mage = 31.74, SDage = 12.13), incorporating 15 females and eight males.”

The study aimed for 55 people, but only got 37 people, with 14 in the positive writing group (10F) and 23 in the neutral control group (15F). Thus, the study focused on 14 people, mostly female, which is much too small.

Author Response

Reviewer comment: In response to my review below (and perhaps other reviews), the authors have added that this is a pilot study. However, in most cases pilot studies are not published; they lead to larger-scale studies which are published.  Because the sample here is so small and so biased (female, young) there is a high probability that the results will not be replicable.These data will confuse/mislead the field, not help the field.

Response to reviewer comment: We agree with the reviewer that it is possible that pilot study findings won't be replicated, and if taken as conclusive findings they have the potential to mislead the field. However, it was for this reason that we clearly refer to our study as a pilot study throughout the Discussion. We respectfully disagree that in most cases pilot studies are not published - there are a large number of pilot studies which are published in the field, including in Nutrients. In fact, it is our view that published pilot studies can in fact advance the field, by prompting future replication attempts. We have modified the conclusing paragraph of the manuscript to further highlight that replication is needed in larger scale studies before conclusive inference can be drawn from this evidence.